# U SO CARE—The Impact of Cardiac Ultrasound during Cardiopulmonary Resuscitation: A Prospective Randomized Simulator-Based Trial

**DOI:** 10.3390/jcm10225218

**Published:** 2021-11-09

**Authors:** Karim Zöllner, Timur Sellmann, Dietmar Wetzchewald, Heidrun Schwager, Corvin Cleff, Serge C. Thal, Stephan Marsch

**Affiliations:** 1Institution for Emergency Medicine, 59755 Arnsberg, Germany; karim.zoellner@yahoo.de (K.Z.); dietmar.wetzchewald@aim-arnsberg.de (D.W.); heidrun.schwager@aim-arnsberg.de (H.S.); 2Department of Anaesthesiology and Intensive Care Medicine, Bethesda Hospital, 47053 Duisburg, Germany; 3Department of Anaesthesiology 1, Witten/Herdecke University, 58455 Witten, Germany; serge.thal@uni-wh.de; 4Department of Anaesthesiology and Intensive Care Medicine, University of Cologne, 50923 Cologne, Germany; corvin.cleff@uk-koeln.de; 5Department of Intensive Care, University Hospital, 4031 Basel, Switzerland; stephan.marsch@usb.ch

**Keywords:** cardiopulmonary resuscitation, echocardiography, simulation, randomized controlled trial, POCUS

## Abstract

Background: Actual cardiopulmonary resuscitation (CPR) guidelines recommend point-of-care ultrasound (POCUS); however, data on POCUS during CPR are sparse and conflicting. This randomized trial investigated the effects of POCUS during CPR on team performance and diagnostic accuracy. Methods: Intensive Care and Emergency Medicine residents performed CPR with or without available POCUS in simulated cardiac arrests. The primary endpoint was hands-on time. Data analysis was performed using video recordings. Results: Hands-on time was 89% (87–91) in the POCUS and 92% (89–94) in the control group (difference 3, 95% CI for difference 2–4, *p* < 0.001). POCUS teams had delayed defibrillator attachments (33 vs. 26 sec, *p* = 0.017) and first rhythm analysis (74 vs. 52 sec, *p* = 0.001). Available POCUS was used in 71%. Of the POCUS teams, 53 stated a POCUS-derived diagnosis, with 49 being correct and 42 followed by a correct treatment decision. Four teams made a wrong diagnosis and two made an inappropriate treatment decision. Conclusions: POCUS during CPR resulted in lower hands-on times and delayed rhythm analysis. Correct POCUS diagnoses occurred in 52%, correct treatment decisions in 44%, and inappropriate treatment decisions in 2%. Training on POCUS during CPR should focus on diagnostic accuracy and maintenance of high-quality CPR.

## 1. Introduction

Guidelines for advanced cardiac life support (ACLS) recommend the use of point-of-care ultrasound (POCUS) during cardiopulmonary resuscitation (CPR) [1,2]. These recommendations are mainly based on beneficial diagnostic effects and/or help to terminate futile CPR. However, in one of the largest reviews to date, the evidence for using POCUS as a prognostic tool for clinical outcomes during cardiac arrest is of very low certainty and is hampered by multiple risks of bias. In the review in question, no sonographic finding had sufficient and/or consistent sensitivity for any clinical outcome to be used as the sole criterion to terminate resuscitation [3].

Theoretically, the robust beneficial diagnostic and therapeutic effects of POCUS during CPR should be available, e.g., in the case of pulmonary embolism (PE) or pericardial tamponade (PT). There are, however, no larger randomized trials on this topic, and observational data available are conflicting and cover the range from no perceived additional benefit to significant alternations of diagnosis and management during pre-hospital care [3,4]. CPR is a stressful task [5] and the availability of POCUS may add supplementary cognitive and emotional demands on CPR teams. Indeed, observational studies have reported an association of POCUS with increased no-flow times [6,7]. Beyond that, not every rescuer is trained in POCUS and might therefore be unable to seize the opportunity to gain additional diagnostic information. Studies in real cases and simulated scenarios have repeatedly shown a high variance in executing CPR and a less than optimal adherence to treatment algorithms [8,9,10,11,12]. Moreover, teams under stress tend to focus on subtasks differently and may even neglect certain subtasks [13,14]. As a result, POCUS during CPR may, by leading to distraction or stress in the rescuers, further impact on variance in executing CPR and/or affect specific subtasks. As mentioned above, two studies have already shown prolonged pauses and delays during CPR when POCUS was available [6,7]. Finally, there are no data on whether leadership is able to mitigate the effects of additional stress, as may be caused by POCUS, despite this being an important factor in CPR teams [15].

Investigating the impact of POCUS on the quality of CPR in a large randomized controlled trial is difficult in real cases, as both the circumstances of arrests and the availability of POCUS may vary substantially. Out of hospital arrests are frequently treated by rescuers not trained in POCUS, making the inclusion criteria in such trials challenging. Moreover, important deviations, delays, and diagnostic errors can occur during the use of POCUS, which can only be captured if recording equipment is functional or trained observers are present at the scene. Simulation allows the investigation of team performances both globally and in specific subtasks in a realistic and standardized manner [16]. A particular advantage of simulation is the possibility of recording data throughout the entire process. Accordingly, the aim of the present prospective randomized trial was to investigate the effects of POCUS on the quality of CPR, the diagnostic accuracy of POCUS, and the effects of designated leadership in simulated cardiac arrests in a mixed population of internal, surgical, and anesthesiologic residents.

## 2. Materials and Methods

### 2.1. Participants

The Working Group on Intensive Care Medicine, Arnsberg, Germany (http://www.aim-arnsberg.de, accessed on 13 September 2021), organizes educational courses for physicians, mainly residents in their 2nd to 3rd year of postgraduate medical education in internal medicine, anesthesia or surgery, from Germany and German-speaking countries working in Intensive Care and Emergency Medicine. Participants of the courses were invited to attend voluntary simulator-based CPR workshops and informed that simulations were video-taped for scientific reasons. Identical workshops were offered to physicians wishing to participate without being filmed. The trial was carried out following the rules of the Declaration of Helsinki and was approved by the Ethics Committee of Aerztekammer Westfalen-Lippe (2017-734-f-S), who waived the obligation to obtain consent. The study was registered at the German Clinical Trial Registry (www.drks.de, accessed on 4 November 2021; DRKS-ID: DRKS00013786) and reported herein according to the extensions to the CONSORT statements of the Reporting Guidelines for Health Care Simulation Research [17].

### 2.2. Study Design

This is a prospective randomized single-blind trial. Randomizations were carried out using computer-generated numbers. Participants from single workshops were randomly assigned to teams of three to five physicians. Teams were then randomly allocated to perform CPR under two different conditions: (1) no POCUS (control group) or (2) POCUS available (POCUS group). Furthermore, teams were randomly allocated (1:1:1) to no designated leadership (no intervention); designated leadership by team itself (team was given the task to designate a leader prior to the start of the scenario); or designated leadership by tutor (leader was assigned to a randomly chosen team member by the tutor prior to the start of the scenario). Designated leaders wore a colored vest and could thus be identified on video recordings. Apart from the availability of POCUS and the assigned leadership, conditions and circumstances for all teams were identical.

### 2.3. Simulator and Scenario

The mannequin Ambu Man Wireless (Ambu GmbH, Bad Nauheim, Germany) was used. All participants received a standardized introduction to the workshop, the mannequins, and the resuscitation equipment available. Subsequently, all team members were informed that their role during the following scenario was that of a resuscitation team summoned to an in-hospital cardiac arrest. The victim of the arrest (mannequin), already resuscitated for 10 min by a bystander (tutor) in a remote hospital area, was handed over to the team. A bradycardic heart rhythm with broadened QRS-complex was displayed on the manual defibrillator simulator (ALSI iSimulate, iSimulate, LLC, Albany, NY, USA). Apart from a wristband identifying the victim as the patient, no further medical information was available. The study period started with the first touch of the patient by one of the participants and ended after 8 min. After handing over the victim, tutors, who were instructed to refrain from any intervention until the end of the study period, operated the resuscitation mannequins. The further course of the scenario was at the discretion of the tutor, who, after the simulated scenario, gave educational feedback to the teams.

### 2.4. Point-of-Care Ultrasound (POCUS)

Teams randomized to the POCUS group received an ultrasound device dummy consisting of a laptop with an attached sector probe for cardiac ultrasound (Figure 1). According to prior randomization, each team had the chance to see one of two loops with unambiguous pathologic echocardiographic findings, derived from prior recordings (pericardial tamponade, PT, or pulmonary embolism, PE), both mentioned in the actual ERC guidelines in the context of POCUS [18]. These loops were stored within a power point presentation (Microsoft, Redmond, WA, USA) and the presentation was started remotely by the tutor if the investigator placed the probe in the subcostal position following the FEEL-protocol (focused echocardiographic evaluation in life support) previously described by Breitkreutz et al. [4]. The two loops are available as electronic Appendix A.

A conventional laptop placed in a bag mimicked the POCUS device. The echo probe was attached in a side pocket. Power point presentations displaying one of two echo loops were started remotely when the echo probe was placed in a subcostal view.

### 2.5. Data Analysis

Data analysis was performed using video recordings obtained during simulations. The first touch of the patient by one of the participants was defined as the starting point for the timing of all events.

The following definitions were used to describe the handling of POCUS: interaction = any device-related hands-on activity; use of POCUS = placing an ultrasound probe in subxiphoid or parasternal position with the device turned on; discussion = any utterance relating to interpretation of POCUS findings; diagnosis = explicit statement of a POCUS-derived diagnosis; correct diagnosis = statement of PE OR right ventricular dilatation (teams randomized to PE loop) and PT (teams randomized to PT loop), respectively; treatment decision = explicit statement of a POCUS-derived treatment decision; appropriate treatment = thrombolysis OR no specific treatment (teams randomized to PE loop) and pericardiocentesis (teams randomized to PT loop), respectively; inappropriate treatment = all treatment decisions not fulfilling the above definition of appropriateness.

### 2.6. Statistical Analysis

The primary endpoint was the percentage of hands-on time, defined as time of actual chest compressions divided by the total time interval of the study period. A power analysis, based on data of pilot experiments, revealed that approximately 90 teams had to be studied in each study arm to detect a between-group difference of 10% in the primary outcome with significance levels of 0.05 and 80% power. Accordingly, we decided to terminate the study as soon as at least 90 videotapes of sufficient quality for each study arm were available. For organizational reasons, the number of available videotapes of sufficient quality could be assessed only after completion of each educational course.

Secondary outcomes included correct use and correct diagnoses of POCUS findings, overall time of interactions with the echocardiographic device, and adherence to various aspects of international CPR guidelines. The effect of designated leadership was also assessed as a secondary outcome.

All data were analyzed on an intention-to-treat basis. Data are expressed as medians [IQR] unless otherwise stated. Statistical analysis was performed using SPSS (version 25). Numerical data were analyzed by non-parametric ANOVA, followed by the Mann-Whitney test, if appropriate. Estimates for differences between medians and their approximate confidence intervals were obtained by the Hodges-Lehmann estimation. Categorical data were analyzed using the chi-square test. A *p* < 0.05 (two-tailed) was considered to represent statistical significance.

## 3. Results

### 3.1. Participants

The data from 191 teams with 668 participants (95 control group; 96 POCUS group) were finally analyzed (CONSORT flow chart, Figure 2). The gender of leaders was equally distributed between control groups and POCUS groups (*p* = 0.96) as wells as between pathologies in POCUS groups (*p* = 0.66).

### 3.2. Primary Outcome

In the intention-to treat analysis, hands-on time was 92% (89–94) in the control group and 89% (87–91) in the POCUS group (difference 3, 95% CI for difference 2–4, *p* < 0.001; Figure 3). A per-protocol analysis comparing the teams not using POCUS (*n* = 124) with the teams using POCUS (*n* = 67) revealed a very similar result (91% (89–94) vs. 89% (86–90); difference 3, 95% CI for difference 2–4, *p* < 0.001). Leadership allocation had no effect on hands-on times (*p* = 0.99).

### 3.3. Secondary Outcomes

Secondary outcomes are displayed in Table 1. Per-protocol analyses comparing teams using and teams not using POCUS revealed very similar results to the intention to treat data shown in Table 1. Designated leadership had no significant effect on any secondary outcome.

A flow chart on POCUS use among the POCUS group is displayed in Figure 4. Out of 96 teams, 74 (77%) interacted with their ultrasound device during 88 (65–113) sec and 67/96 teams (71%) used POCUS for 24 (14–41) sec (Figure 3). Overall, 2/96 teams (2%) with available POCUS or 2/67 teams (3%) using POCUS took an inappropriate treatment decision (one omission of pericardiocentesis; one pericardiocentesis without indication).

Availability of POCUS thus resulted in 52% (49/96) correct diagnoses, 44% (42/96) correct treatment decisions, and 2% (2/96) inappropriate treatment decisions. Diagnostic accuracy and appropriateness of the treatment decision was not affected by the allocated cardiac pathology nor by the leadership allocation.

## 4. Discussion

This prospective randomized simulator-based trial demonstrated that the availability of POCUS during CPR was associated with lower hands-on times and delays in time to first rhythm analysis. Available POCUS was used by approximately 70% of teams and overall resulted in 44% correct and 2% inappropriate treatment decisions, respectively.

A key feature of the present study is that it was designed to identify the status quo of POCUS during CPR and therefore did not contain any previous study-driven training in POCUS use. In prior work, POCUS procedures were often performed in protocolized ways, with significant attention given to not interrupting CPR or lengthening pulse checks, as well as to training in what findings to look for that might alter treatment [19].

To the best of our knowledge, the present study is the first large-scale randomized trial of the effects and diagnostic accuracy of POCUS on the quality of CPR. Our findings confirm prior observational research demonstrating that POCUS is associated with lower “hands-on times” as evidenced by delays in chest compressions [7] or prolonged “no-flow time” [6]. Accessing “no-flow time” associated with POCUS use during CPR in one study found that pauses were <10 sec in only 44% (4/9) cases, with a median pause length of 17 sec [20]. Whereas prolongation of no-flow times has been described before [6,7,20], our study is the first to show that in addition to hands-on times further key components of CPR, such as defibrillator attachment, time to first rhythm analysis, and advanced airway management were delayed in the POCUS group.

POCUS has attained a safe standing in emergency and critical care medicine as well as in other fields of medicine. For its use in the emergency department setting, the American College of Emergency Physicians recently identified five main areas of POCUS scope including resuscitation, diagnosis, procedural guidance, signs/symptom evaluation, and therapeutic or monitoring indications [21]. In one of the latest reviews on this topic, Long et al. described POCUS as a valuable diagnostic and prognostic tool in cardiac arrest, with the recent literature supporting its diagnostic ability [22]. However, there are also concerns about terminating resuscitation on the basis of POCUS findings alone, precisely because of diagnostic inaccuracies and potential misinterpretation [23]. To date, data on misdiagnosis and inappropriate treatment associated with POCUS during CPR are limited [24].

Alongside its use during CPR, an acceptable diagnostic accuracy is described for the evaluation of acute dyspnea in the emergency department [25], lung ultrasound during COVID-19 [26,27], ward emergencies [28], or for use as an adjunct to the reexamination of physical findings [29].

Potential misdiagnoses include pericardial effusion vs. pleural vs. ascites vs. epicardial fat, as well as right ventricle dilation in acute pulmonary embolism and inferior vena cava for volume status assessment in cardiac ultrasound or lung point and lung pulse misinterpretations and mirror artifacts vs. lung consolidations in lung ultrasound. 

The authors from this study conclude that by integrating POCUS findings into a broader clinical context, most POCUS misdiagnosis can be avoided, and thus patients’ safety can be enhanced [28].

In our study, diagnostic accuracy of POCUS was around 85% with an incidence of inappropriate treatment decisions of 2%, an observation that has not been reflected in any major work to date. Overall, the literature on this topic is characterized by small observational studies and case series, limiting the conclusions that can be drawn [19] and justifying the execution of further studies in this field.

As course participants were not exclusively cardiologists, but derived from surgery, anesthesiology, and general internal medicine, we tried to select echo loops that were as unambiguous as possible from disease patterns that are explicitly mentioned in the current guidelines. Still, there remained a chance of multiple missed findings.

The actual ERC ACLS guideline recognizes the increasing role of POCUS in peri-arrest care for diagnosis, but emphasizes that it requires a skilled operator, and that there is a need to minimize interruptions during chest compression [18]. So far, data on the actual use of POCUS during CPR in the real world are sparse and the present study aims to contribute to the determination of usefulness of POCUS during CPR. Moreover, data on the overall usefulness of POCUS during CPR are conflicting and cover the range between perceived benefit by enhancement of diagnosis leading to adaption and specification of management and no perceived additional benefit [3,4,18].

Another potential barrier for an extensive introduction of POCUS during CPR may be time delays imposed by device start up times in one- or two-person resuscitation. The time to establish operational readiness varies between 2 and 3 sec if the devices are in stand-by up to 90 sec for a cold boot.

Our study has several implications: First, Honarmand [30] and Crowley [31] have both highlighted the importance of adherence to ACLS protocols and that deviation from ACLS guidelines were associated with a lower likelihood of ROSC and survival to hospital discharge. Other than delays in CPR, divergence from airway management as guideline-recommended or incorrect time for pulse/rhythm check was identified as a major deviation influencing return of spontaneous circulation [30,31]. Thus, by extrapolating the combination of shortcomings in CPR associated with POCUS use in our study to these real-world data, at least a moderate effect on outcomes may be expected. Second, there are a limited number of echocardiographic diagnoses during CPR that are relevant in the sense that they lead to specific therapeutic interventions (PT or pneumothorax), whereas others are not necessarily accompanied by a change in therapy (right ventricular dilation in isolation as seen during PE or PEA). To improve diagnostic accuracy, training in POCUS should focus on relevant conditions. In addition, analogous to the 4Hs and 4Ts, CPR guidelines could summarize the relevant diagnoses that should be confirmed or excluded by POCUS [18]. Third, potential rescuers should be made aware that the use of POCUS during CPR may result in less than optimal CPR quality and of their duty to prevent this from happening. If POCUS is declared to be an integral part of CPR, teaching and training of rescuers must consider the following two factors independently but also interdependently: not only does ACLS have to be trained, but also POCUS and its use during ACLS.

The strengths of this trial include the large sample size and the perfectly identical conditions for all teams. The limitations of simulator-based studies include the absence of real patients and, in the present trial, of real POCUS assessments. However, simulation is increasingly regarded as an accepted tool for evaluation [16], while performance markers in simulator-based studies show a high agreement with findings in real cases.

In order to assess the actual state in training and benefits of POCUS, we refrained from using special POCUS protocols, or prior teaching such as workshops, didactics, phantom, or live model training. Every resident participated in the study only once and it may be possible that longer study times would have resulted in more frequent POCUS use, as would probably have been the case with habituation. In an attempt to observe real-world data, we chose a study population consisting of 2nd to 3rd year residents who, at the time of the study, were serving as potential first responders in their hospitals.

There are currently no standardized guidelines for simulation in POCUS training and the current literature on POCUS simulation as a training tool is limited [32]. A pitfall of POCUS simulation is that it may reproduce procedural training [33]. Although special ultrasound dummies do exist, to our knowledge, there is no commercially available trainer enabling POCUS, especially during CPR; for this reason, our tutors remotely started the randomized echocardiographic loops once the ultrasound probe was placed as recommended by the FEEL protocol introduced by Breitkreutz and colleagues [4]. As displayed in CONSORT flow chart, this test sequence worked well for 99% (96/97) of the teams.

## 5. Conclusions

The availability of POCUS during CPR resulted in lower hands-on times, lower performance in additional CPR benchmarks, correct diagnoses in 52% of cases, correct treatment decisions in 44%, and inappropriate treatment decisions in 2%. Training on POCUS during CPR should focus on high diagnostic accuracy and the maintenance of high-quality CPR.

## Figures and Tables

**Figure 1 jcm-10-05218-f001:**
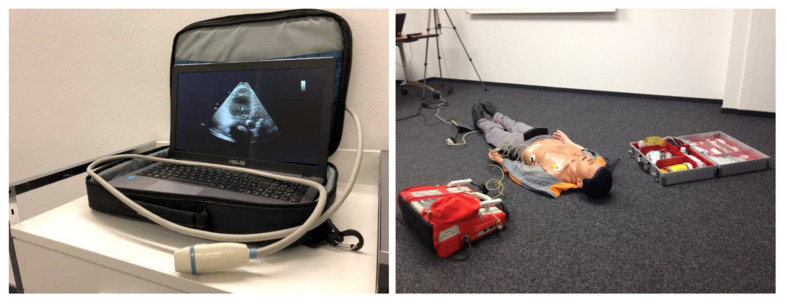
Test setup.

**Figure 2 jcm-10-05218-f002:**
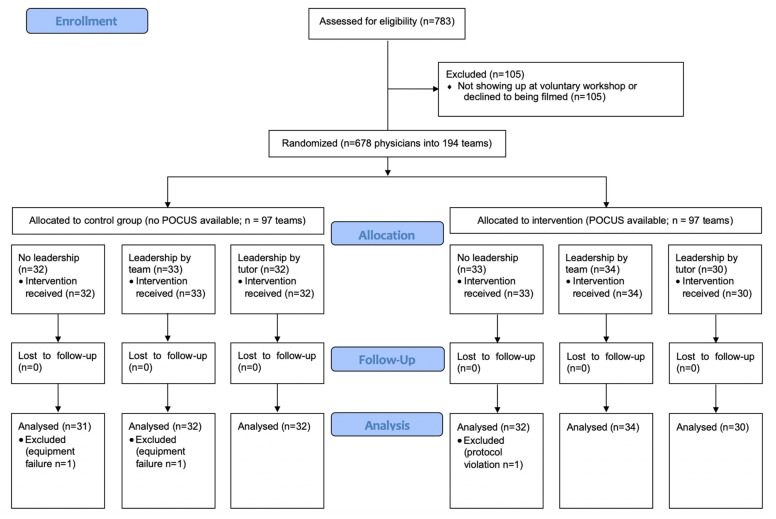
CONSORT flow chart.

**Figure 3 jcm-10-05218-f003:**
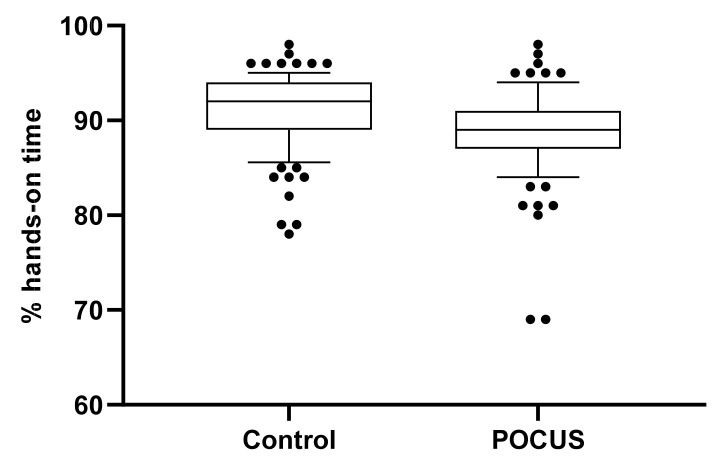
Box and whisker plot of the percentage hands-on time. Boxes represent medians and interquartile range; whiskers delineate the 10th and 90th percentile respectively. Dots indicate values outside the 10th and 90th percentile. Left bar = control group (no POCUS available); right bar = intervention group with POCUS available. Overall hands-on times differed significantly between the groups (*p* < 0.001).

**Figure 4 jcm-10-05218-f004:**
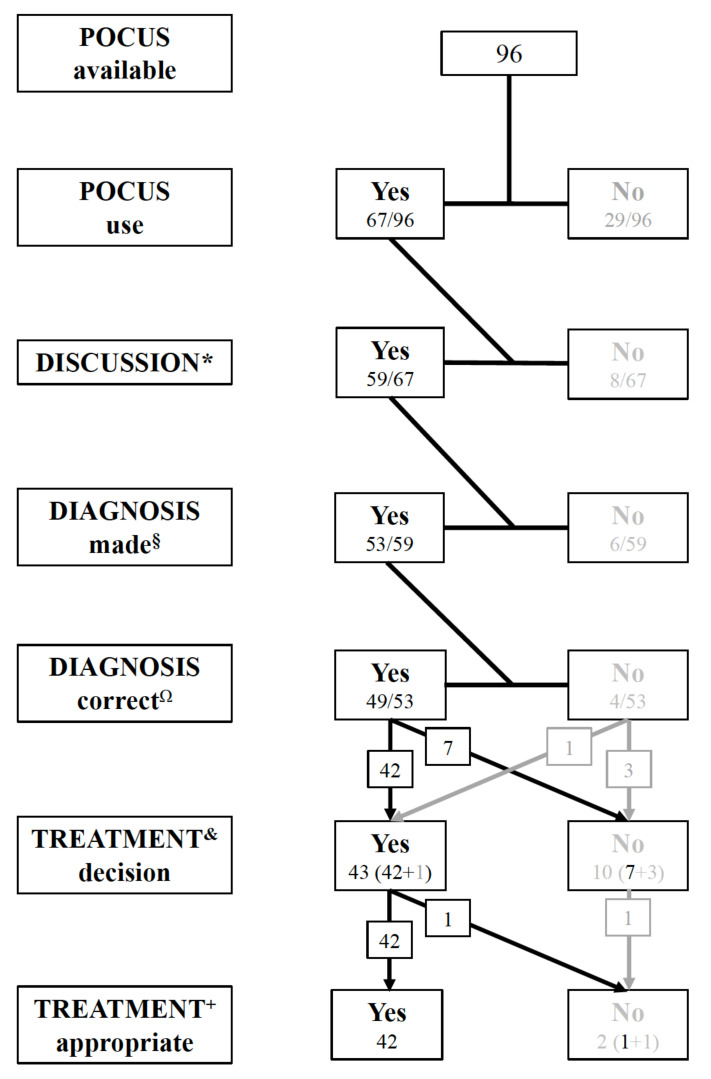
Flow chart on POCUS use among the 96 teams of the POCUS group: * Discussion = any utterance/statements relating to POCUS interpretation; ^§^ Diagnosis made = explicit statement of a POCUS-derived diagnosis; ^Ω^ Diagnosis correct = statement of pulmonary embolism OR right ventricular dilatation (teams randomized to embolism loop) and pericardial tamponade (teams randomized to tamponade loop); ^&^ Treatment decision = explicit statement of a treatment decision; ^+^ Treatment appropriate = thrombolysis OR no specific treatment (teams randomized to PE loop) and pericardiocentesis (teams randomized to PT loop), respectively; Inappropriate treatment sub summarized all treatment decisions not fulfilling the above definition of appropriateness.

**Table 1 jcm-10-05218-t001:** Secondary outcomes.

	Control Group (*n* = 95)	POCUS Group (*n* = 96)	*p*
Start cardiac massage (sec)	6 (6–10)	6 (6–10)	0.11
Chest compression rate (strokes/min)	108 (100–113)	104 (97–113)	0.053
Start ventilation (sec)	73 (44–109)	66 (46–112)	0.85
AAM completed (sec)	96 (59–147)	110 (69–168)	0.10
SGA:ETI	51:44	54:41	0.77
Ventilatory rate (breaths/min)	11 (8–18)	11 (7–15)	0.24
Defibrillator attached (sec)	26 (16–39)	33 (20–56)	0.017
1st rhythm analysis (sec)	52 (36–79)	74 (45–109)	0.001
Erroneous shock delivered (teams)	10	16	0.29
i.v. line inserted (sec)	87 (66–124)	91 (65–131)	0.52
Epinephrine administered (sec)	189 (150–254)	185 (144–249)	0.99

Data are medians (IQR). AAM = advanced airway management; SGA = supraglottic airway; ETI = endotracheal intubation.

## Data Availability

The data presented in this study are available on request from the corresponding author.

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
