# Peer review of "U SO CARE—The Impact of Cardiac Ultrasound during Cardiopulmonary Resuscitation: A Prospective Randomized Simulator-Based Trial"

_jcm, 2021, doi:10.3390/jcm10225218_

Round 1
Reviewer 1 Report
the authors seem to propose a simulation model for assessment of pocus findings during cardiac arrest can offer insights into implementation real world cardiac resuscitation
the authors thoughtfully touched on the context of usefulness of pocus depending on underlying clinical considerations and the benefit of focusing on specific clinical questions to ask of pocus in the setting of resuscitation
the discussion might be strengthened by addressing the context of clinical history in the usefulness of pocus and the setting in which resuscitation is being offered (in hospital vs out of hospital) (burden of patient morbidity, preexisting conditions that have characteristic mechanisms for leading to cardiac arrest)
the work is important in furthering the scientific conversation about the use and incorporation of pocus in standard clinical care
randomized studies are of limited value if they include a binary use or non-use of pocus as the authors note user experience and clinical context define the case by case benefit of pocus utility and in cases where pocus defined a clinical treatment plan, the alternative non-pocus intervention cannot be compared
cardiac arrest is inherently the final common event for all humans and is an inevitability therefore defining context and clinical goals is central to leveraging pocus or any resuscitation effort in determining its utility and measuring an endpoint
additionally, I feel it may be useful for the authors to mention device start up time as a potential barrier to timely pocus use and in 1 person or 2 person resuscitation there may be limited utility in even attempting pocus
I would like to congratulate the authors on their efforts to design and implement this research as it is a meaningful stepping stone in furthering the pocus conversation
Reviewer 2 Report
General Comments:
In this original manuscript entitled, "The impact of cardiac ultrasound during cardiopulmonary resuscitation: A prospective randomized simulator based trial", Zollner et al. reviewed timely and important topic. I commend the authors for their choice of topic and manuscript. I have a few comments below for their consideration:
- Though POCUS is a very important tool in clinical practice, it is important to understand that people interpreting these echocardiograms are not cardiologists. Often there are multiple missed findings.
- Furthermore, ultrasonography of the lungs and abdomen/pelvis may be useful in addition to cardiac ultrasound to assess for alternate etiologies of cardiac arrest. The authors need to discuss these issues in greater detail.
- Important debriefing qualitative analysis regarding not using POCUS despite its availability is crucial. This will help us understand better the rationale of treating physicians to use or not use this technology despite it being present.
- I would asked the authors to elaborate what specific findings/diagnoses that they found on this technology that helped him change management. Outside of pericardial effusion, pulmonary embolism or severe valvular heart disease, I am unsure what additional information from a structural/hemodynamic standpoint that point-of-care ultrasonography presents.
- The risk of wrong intervention specifically, mixing up epicardial fat for pericardial effusion, tapping pleural effusion for a pericardial effusion or confusing post-arrest myocardial dysfunction for acute RV dysfunction from pulmonary embolism or significant confounders. Therefore, I strongly believe that echocardiography in such patients should be performed by qualified cardiologist/echocardiography as as against ultrasound enthusiasts.
Round 2
Reviewer 2 Report
None